# The Effect of Diffuse Liver Diseases on the Occurrence of Liver Metastases in Cancer Patients: A Systematic Review and Meta-Analysis

**DOI:** 10.3390/cancers13092246

**Published:** 2021-05-07

**Authors:** Filippo Monelli, Giulia Besutti, Olivera Djuric, Laura Bonvicini, Roberto Farì, Stefano Bonfatti, Guido Ligabue, Maria Chiara Bassi, Angela Damato, Candida Bonelli, Carmine Pinto, Pierpaolo Pattacini, Paolo Giorgi Rossi

**Affiliations:** 1Clinical and Experimental Medicine PhD Program, University of Modena and Reggio Emilia, 41124 Modena, Italy; filippo.monelli@ausl.re.it; 2Radiology Unit, Department of Diagnostic Imaging and Laboratory Medicine, AUSL-IRCCS di Reggio Emilia, 42123 Reggio Emilia, Italy; pierpaolo.pattacini@ausl.re.it; 3Epidemiology Unit, AUSL- IRCCS di Reggio Emilia, 42123 Reggio Emilia, Italy; olivera.djuric@ausl.re.it (O.D.); laura.bonvicini@ausl.re.it (L.B.); paolo.giorgirossi@ausl.re.it (P.G.R.); 4Center for Environmental, Nutritional and Genetic Epidemiology (CREAGEN), Section of Public Health, Department of Biomedical, Metabolic and Neural Sciences, University of Modena and Reggio Emilia, 41124 Modena, Italy; 5Radiology Unit, AOU Policlinico di Modena, University of Modena and Reggio Emilia, 41124 Modena, Italy; roberto.fari2@gmail.com (R.F.); bonfste@gmail.com (S.B.); guido.ligabue@unimore.it (G.L.); 6Medical Library, AUSL- IRCCS di Reggio Emilia, 42123 Reggio Emilia, Italy; mariachiara.bassi@ausl.re.it; 7Oncology Department, AUSL- IRCCS di Reggio Emilia, 42123 Reggio Emilia, Italy; Angela.damato@ausl.re.it (A.D.); candida.bonelli@ausl.re.it (C.B.); carmine.pinto@ausl.re.it (C.P.)

**Keywords:** diffuse liver disease, NAFLD, hepatic viral infection, cirrhosis, liver metastasis, colorectal cancer, lung cancer, breast cancer, pancreatic cancer

## Abstract

**Simple Summary:**

Diffuse liver diseases have a high incidence among the general population and even higher in patients with a solid cancer. Since many patients with a solid tumor die of liver metastases, the aim of this systematic review of the literature was to explore the correlation between diffuse liver diseases and the risk of having liver metastases at diagnosis or during follow-up. To summarize the results of included studies, a meta-analysis was also conducted. The results of our systematic review should encourage the research community to further investigate the complex relationship between the liver’s microscopic environment and metastases, which may also affect prognosis and response to therapy.

**Abstract:**

This systematic review with meta-analysis aimed to assess the effect of diffuse liver diseases (DLD) on the risk of synchronous (S-) or metachronous (M-) liver metastases (LMs) in patients with solid neoplasms. Relevant databases were searched for systematic reviews and cross-sectional or cohort studies published since 1990 comparing the risk of LMs in patients with and without DLD (steatosis, viral hepatitis, cirrhosis, fibrosis) in non-liver solid cancer patients. Outcomes were prevalence of S-LMs, cumulative risk of M-LMs and LM-free survival. Risk of bias (ROB) was assessed using the Newcastle-Ottawa Scale. We report the pooled relative risks (RR) for S-LMs and hazard ratios (HR) for M-LMs. Subgroup analyses included DLD, primary site and continent. Nineteen studies were included (*n* = 37,591 patients), the majority on colorectal cancer. ROB appraisal results were mixed. Patients with DLD had a lower risk of S-LMs (RR 0.50, 95% CI 0.34–0.76), with a higher effect for cirrhosis and a slightly higher risk of M-LMs (HR 1.11 95% CI, 1.03–1.19), despite a lower risk of M-LMs in patients with vs without viral hepatitis (HR 0.57, 95% CI 0.40–0.82). There may have been a publication bias in favor of studies reporting a lower risk for patients with DLD. DLD are protective against S-LMs and slightly protective against M-LMs for viral hepatitis only.

## 1. Introduction

In most cases, cancer causes death through the growth of distant metastases in vital organs [1]. Due to its double venous system and complex lymphatic system, the liver is one of the most frequent sites of distant metastases. This is particularly the case for colon cancer, with half of associated metastases in the liver; rectal, breast, pancreatic and lung cancers also frequently metastasize to this organ. Thus, metastatic disease frequently affects prognosis [2], management and therapeutic choices [3].

Diffuse liver diseases include various pathological conditions, some of which, such as liver steatosis, fibrosis and cirrhosis, are very common worldwide. In fact, steatosis is the hallmark of non-alcoholic fatty liver disease (NAFLD), which has an overall global prevalence of 25% [4], while fibrosis and cirrhosis are the final result of several insults to the liver, including chronic viral infections, which reach a prevalence of nearly 5% [5].

Cancer patients have at least the same probability of having diffuse liver disease as the general population, though it is probably higher, given the known association between cancer development and metabolic syndrome, frequently characterized by liver steatosis [6]. Moreover, cancer patients have a higher risk of developing diffuse liver diseases as a consequence of treatment.

In recent years, pathogenetic research and new cancer treatments have focused more on tumor and organ microenvironments than on the cancer cells themselves [7]. Since the microenvironment of the target organ is recognized as a central factor in the process of forming organ metastases, it is reasonable to think that diffuse liver diseases may have an impact on the occurrence of liver metastases. In fact, the ability of metastatic cells to survive and proliferate in the liver is determined by the complex interactions between tumor cells and preexisting tissue cells, including the sinusoidal endothelium, stellate, Kupffer and inflammatory cells [8]. This interplay may be modified by diffuse liver diseases, which influence the liver microenvironment, thereby potentially favoring or hindering the development of hepatic metastases [9,10]. Similarly, chemotherapy-induced liver modifications, although temporary in some cases [11], may also influence the probability of developing liver metastases.

Some studies have analyzed the association between the presence of diffuse liver diseases and liver metastasis occurrence in cancer patients, but the literature on this topic appears to be fragmented and somewhat contradictory.

The aim of this systematic review was to assess the effect of the presence of diffuse liver diseases (steatosis, viral hepatitis, cirrhosis and fibrosis) on the risk of having liver metastasis at diagnosis (synchronous) or developing liver metastases after the diagnosis (metachronous) in patients with solid neoplasms, excluding hepatic primary tumors.

## 2. Materials and Methods

### 2.1. Study Eligibility

Systematic reviews and cross-sectional or cohort studies were eligible if they assessed whether the risk and timing of developing liver metastases in patients with solid cancers was different between patients with and those without chronic liver injury. Studies specifically addressing the recurrence of liver metastases after an R0 (i.e., without residual disease) liver resection for metastases of solid neoplasms in patients with and without liver injury at the moment of resection were included. The complete protocol has been registered in the PROSPERO database (ID CRD42019133519).

The diffuse liver diseases considered were liver steatosis, chronic viral hepatitis or chronic hepatitis virus infection, liver cirrhosis and liver fibrosis. The primary tumors considered were all solid tumors except primary liver neoplasms.

The included studies had to have reported a direct comparison between two groups of patients with a solid neoplasm: patients with chronic liver injury (exposed) and patients without chronic liver injury (nonexposed).

Exclusion criteria were hematologic neoplasms, primary liver neoplasms, absence of a comparison between different exposure levels, animal studies, studies on solid malignancies with a follow-up of less than 12 months in cases of lung and pancreatic cancer or 24 months for every other type of tumor. Studies reported only as abstracts or published in languages other than English, German, Spanish, French and Italian were excluded.

### 2.2. Outcomes and Rationale of the Comparisons

The primary endpoints considered were the presence of liver metastases at diagnosis (synchronous) and the development of liver metastases during follow-up (metachronous). For synchronous metastases, the outcome was computed as prevalence of cases with liver metastases out of the total number of diagnosed cases; for metachronous metastases, the outcomes considered were liver metastasis-free survival (measured as hazard ratio from survival times) and cumulative risk of liver metastases (measured as the proportion after a predefined follow-up or rate using person/time as denominator). Overall survival was considered only for the studies reporting liver metastasis-free survival as well. Finally, the overall cumulative risk of liver metastases (synchronous and metachronous together) was considered for studies in which both types of metastases were explicitly reported for all incident cancers occurring in the same population and in the same time period. Overall survival was considered only to determine whether differences in this outcome between patients with and those without liver disease introduced a bias due to competitive mortality that may have affected the accuracy of liver metastasis-free survival. Instead, the cumulative incidence of synchronous and metachronous metastases was included to determine whether differences between patients with and those without liver disease in the prevalence of synchronous metastases could have been due to a difference in the detection of prevalent metastases. If the presence of a liver disease affected the probability of detecting liver metastases at diagnosis because of the different number, type and accuracy of tests performed in patients with or without liver disease, undetected metastases would occur in follow-up as metachronous metastases. Thus, by comparing the sum of metastases occurring at diagnosis and those in follow-up, we should overcome this possible bias, on the condition that the patients included in the two analyses represent an unselected sample of all incident cancers in the same period.

### 2.3. Study Search and Selection

A systematic search was conducted in MEDLINE, The Cochrane Library, EMBASE and Scopus, adapting the search strategy (Appendix B) to the requirement of each database including studies from 1990. This date limit was introduced because of the presence of a certain heterogeneity in the definition of “chronic liver injury,” which is more relevant in studies before 1990 [12]. This is particularly important for chronic viral infection, since hepatitis C virus was discovered in 1989 [13] and the first clinical diagnostic tests were developed in 1990 [14].

The last search was conducted in September 2019; the search strategy designed for MEDLINE is reported in the Supplementary Methods section.

One reviewer (FM) screened the search results based on title/abstract; a second reviewer (GB) screened a computer-generated random sample of 25% of the references to identify potential disagreement, resolved by consensus. Then, two reviewers (FM and GB) independently examined eligibility based on the full text of the relevant articles in a two-step procedure: first, removing articles not pertinent to the research question and second, removing articles without a specific analysis of the eligibility criteria and considered outcomes. In cases of disagreement, inclusion was decided by group consensus involving a third reviewer (PGR).

### 2.4. Data Extraction and Synthesis

Two reviewers (FM and RF) extracted data on study design, country, objective, population (number and characteristics of included patients and controls), how the diagnosis of diffuse liver disease was performed, tumor type, outcomes, inclusion and exclusion criteria, presence, type and length of follow-up and results. Differences between reviewers were resolved by consensus; when this was not possible, by a third reviewer (GB). These data were collected in a standard data extraction form. Two reviewers (FM and SB) used the Newcastle–Ottawa Scale (NOS) [15] to assess selected studies for the risk of bias by consensus; a third reviewer was called in in cases of disagreement (PGR). In the absence of a universally accepted assessment tool for cross-sectional studies, we chose to modify the NOS to evaluate them by removing non-relevant fields. The original and modified NOS are reported in the Appendix A.

### 2.5. Statistical Analysis

Descriptive analysis was done to summarize the distribution of synchronous and metachronous metastases among exposed and non-exposed patients, as well as to summarize the survival among those studies reporting metachronous metastases to assess whether survival bias could have influenced the results.

Separate meta-analyses were performed for synchronous and metachronous metastases. For synchronous metastases, relative risks (RR) were combined and the pooled RRs, with the 95% CI, were calculated, while for the metachronous metastases, hazard ratios (HR), with 95% CI, were calculated. Both types of point estimates were calculated using the random-effects model described by DerSimonian and Laird [16,17]. When available, adjusted HRs obtained in multivariable analyses were used. Missing HRs and standard errors were imputed using methods for incorporating summary time-to-event data into the meta-analysis, as described by Tierney et al. [18].

Subgroup analyses were performed to investigate the risk of metastases by type of liver injury, primary cancer origin and continent where the study was performed. In all analyses which were not divided by type of liver injury, only one liver injury per study was considered, i.e., if one study reported results for more than one type of liver injury, the classification considered for the primary objective of the study was included in the analysis; in the case of no clear definition of the main objective, the liver disease with the highest number of patients was included.

Forest plots were used to display the RRs or HRs and their corresponding 95% CI. Heterogeneity among the studies was evaluated using I2 statistics. Values of I2 can be interpreted as not important (0–40%), moderate (30–60%), substantial (50–90%) and considerable (75–100%) levels of heterogeneity [19]. The possibility of a publication bias was assessed visually using a funnel plot for asymmetry. Meta-analyses were performed using STATA 13.0, metan command.

## 3. Results

### 3.1. Study Selection and Characteristics of the Included Studies

The study selection, according to the PRISMA flow diagram, is reported in Figure 1. For excluded studies, reasons for exclusion are reported in Appendix A.

Nineteen studies met the inclusion criteria and were included in synthesis, for a total of 37,591 patients: 6868 exposed and 29,992 non-exposed (Table 1). Among the 19 included studies [20,21,22,23,24,25,26,27,28,29,30,31,32,33,34,35,36,37,38], five reported overall survival [23,24,31,33,34]. The majority of the selected studies focused on liver metastases from colorectal cancer, while four focused on other cancers. Analyzing the etiology of liver disease, of the studies on colorectal cancer, three are on liver cirrhosis, four on liver steatosis, eight on hepatic viruses and one on liver fibrosis; two studies include two etiologies, analyzing both steatosis and cirrhosis in one case and hepatic viruses in the other. Of the non-colorectal cancer studies, two are on hepatic viruses, pancreatic cancer and nasopharyngeal carcinoma and two on hepatic steatosis, breast cancer and non-small cell lung cancer. Sixteen studies are on first occurrence of metastasis from primitive cancer and three on metastasis recurrence after R0 liver metastasis resection, two on colorectal cancer metastasis to liver steatosis and two on colorectal cancer metastasis to the liver with chronic hepatic virus infection.

It was possible to extract quantitative information on synchronous liver metastases from 12 studies and quantitative information on metachronous liver metastases from 11 studies; four provided information on both metastasis types, but only two of these were studies on all incident cases (Qian 2014 [31], Zeng 2013 [33]), while the other two included patients after liver metastasis resection. Publication dates ranged from 1992 to 2019, but the time span was much shorter for studies addressing metachronous metastases (2012–2019).

### 3.2. Risk of Bias

The results of the consensus appraisal according to NOS and modified NOS are presented in Table 2. The scores attributed to the selection domain were generally high, reflecting a correct selection of exposed and non-exposed cohorts and a clear definition of exposure, while some concern for studies evaluating metachronous liver metastases derived from the uncertain absence of metastases at study start. As for the comparability domain, the major concern was the lack of homogeneity between exposed and non-exposed patients in terms of major confounders, this being truer for hospital-based rather than population-based studies. However, among the included studies on metachronous liver metastases, eight presented adjusted analysis (Chiou 2014 [20], Hamady 2012 [23], Murono 2013 [26], Qian 2014 [31], Kondo 2016 [34], Wu 2017 [35], Wu 2019 [36], Li 2019 [38]) one did not use adjustments but included analysis on matched patients (Ramos 2015 [24]) and only two (Zeng 2013 [33] and Li Destri 2013 [30]) did not provide adjusted hazard ratios. Among confounders included into the adjusted analyses, age, sex, T and N staging components were always present, although two studies did not provide clear indications of considered confounders (Qian 2014 [31] and Chiou 2014 [20]). Finally, the low scores of the outcome domain were frequently due to the inadequate assessment of the occurrence of metastases and the duration of follow-up.

### 3.3. Meta-Analyses

#### 3.3.1. Synchronous Metastases

Meta-analysis of the overall risk of having synchronous metastases in patients with vs without diffuse liver diseases showed lower risk among the former group (RR 0.50; 95% CI 0.34–0.76), with considerable overall heterogeneity (I2 89.4%) (Figure 2). The difference between exposed and non-exposed patients was higher for patients with cirrhosis (RR 0.14; 95% CI 0.07–0.27; I2 0%) than for those with steatosis (RR 0.37; 95% CI 0.15–0.93; I2 80%). Although this result was compatible with random fluctuations, patients with viral hepatitis also had a slightly lower risk of synchronous metastases when compared with patients without viral hepatitis (RR 0.68; 95% CI 0.42–1.10). In this last case, heterogeneity was important (I2 86.9%), with the three largest studies going in the opposite direction (Huo 2018 [28] colorectal cancer; Wei 2013 [37] pancreatic cancer) or showing no effect (Kin Pan Au 2018 [27] colorectal cancer).

When stratified by primary cancer site, the risk of developing synchronous metastases was higher for patients without liver injury if they had colorectal cancer as primary cancer site (RR 0.44; 95% CI 0.29–0.69; I2 88.5%). Only one study reported results on pancreatic cancer as the primary cancer site (Wei 2013 [37]) and showed a higher risk of synchronous metastases for those with liver injury (RR 1.44; 95% CI 1.06–1.95) (Figure 3).

When stratified by the continent where the study was conducted, pooled risk estimates obtained in the studies conducted in and those outside of Asia were similar to the overall estimate (Figure 4). For synchronous metastases, the range of publication dates was wide, with older and smaller studies reporting stronger protective effects than more recent studies.

#### 3.3.2. Metachronous Metastases

Overall, people with diffuse liver diseases had a slightly higher risk of developing metachronous metastases (HR 1.11, 95% CI 1.03–1.19), with considerable overall heterogeneity (I2 78%) (Figure 5). However, the only result with low heterogeneity and showing a more convincing effect was in the opposite direction, i.e., a lower risk in patients with viral hepatitis vs those without (HR 0.51, 95% CI 0.35–0.75; I2 0%). Pooled results for steatosis showed a slight increase in risk among exposed patients, with high heterogeneity (HR 1.12; 95% CI 1.01–1.25; I2 80.9%) and similar results were obtained for cirrhosis (HR 1.15; 95% CI 1.04–1.28) and fibrosis (HR 2.87; 95% CI 1.17–7.03), although estimated in one study only.

When stratifying by cancer site, results were heterogeneous between groups. Pooled analysis was possible only for colorectal cancer (HR 1.12; 95% CI 1.03–1.21; I2 80.8%), while for other cancer sites the risk of metachronous liver metastases in patients with liver disease was either higher (HR 1.43; 95% CI 1.02–2.01 for lung cancer) or lower (HR 0.55; 95% CI 0.35–0.86 for breast). For nasopharyngeal cancer, the study was underpowered and inconclusive (HR 0.77; 95% CI 0.38–1.58) (Figure 6).

When stratified by the continent where the study was conducted, pooled risk estimates obtained in the studies conducted in and outside of Asia were similar to the overall estimate, without significant heterogeneity between groups (Figure 7).

When excluding the two studies which did not provide adjusted HR (Zeng 2013 [33] and Li Destri 2013 [30]), the overall estimate did not change.

#### 3.3.3. Studies Evaluating Synchronous and Metachronous Liver Metastases in the Same Population

Two studies permitted an evaluation of the cumulative incidence of both synchronous and metachronous liver metastases on the same patient groups, with and without viral hepatitis (Qian 2014 [31] and Zeng 2013 [33]). Both showed lower risk of metastases in patients with liver disease: in Qian 2014 [31], cumulatively 9.7% vs. 25.5% and in Zeng 2013 [33], 9.0% vs. 19.6% (Figure 8).

#### 3.3.4. Publication Bias

The funnel plots for both synchronous and metachronous metastases were not symmetrically distributed and several studies were outside of the 95% CI (Figure 9), suggesting a publication bias toward the studies reporting lower risk for patients with diffuse liver disease.

#### 3.3.5. Overall Survival in Studies Reporting Liver Disease Free Survival

Five-year overall survival ranged from 39.3% to 79.8% among the exposed patients and from 39.3% to 92.2% among the non-exposed patients. Two studies (Hamady 2012 [23]; Kondo 2016 [34]) reported lower 5-year survival among exposed patients, while three studies (Ramos 2015 [24]; Qian 2014 [31]; Zeng 2013 [33]) reported higher 5-year survival among non-exposed patients. Given the slight differences, with opposite directions, in overall survival among the studies, compatible with random fluctuations, it is unlikely that differences in overall survival could introduce a serious bias in the results of meta-analyses for liver metastasis survival (Table 3).

## 4. Discussion

The results of this systematic review and meta-analysis show a slight protective effect of diffuse liver diseases on the presence of synchronous liver metastases. Since this protective effect was more apparent for the most severe liver injury (liver cirrhosis), a dose-response relationship can be hypothesized. A slight protective effect was also found for viral hepatitis on metachronous liver metastases, while the presence of other diffuse liver diseases had no effect or resulted in a slight increase in the risk of developing metachronous liver metastases. The results of the included studies were inconsistent. A considerable heterogeneity was found both overall and when stratifying for type of diffuse liver disease and primary cancer site, while no heterogeneity was found when stratifying by the continent where the study was conducted.

Our analyses suggest a strong publication bias, with smaller studies often supporting extreme results particularly in favor of a lower risk of metastases in patients with liver disease. Hospital-based studies may underestimate the risk of metastases in patients with liver disease due to selection bias. In fact, research hospitals can attract more complex cases and complexity could be due to a preexisting condition such as liver disease or to cancer severity. Thus, in hospital-based rather than population-based studies, patients with diffuse liver diseases could be centralized to regional research hospitals at less advanced cancer stages than are patients without liver disease.

The high inconsistency and heterogeneity may partially derive from the design and patient inclusion criteria applied by most of the studies found in the literature. In fact, most studies assessing metachronous metastases did not also evaluate synchronous metastases. By excluding patients with baseline liver metastases, the risk is to look at one part of the picture only. For instance, patients with a diffuse liver disease may undergo a higher number and different types of diagnostic examinations. This may lead to anticipating the diagnosis of a primary tumor, thereby shifting stage at diagnosis towards non-metastatic primary tumor in exposed patients. This early diagnosis may explain the finding of a protective effect of cirrhosis on synchronous but not on metachronous liver metastases. Further, this protective effect is seen in colorectal cancer patients, who may benefit from earlier cancer detection and not in pancreatic cancer patients, seemingly confirming the hypothesis of an early diagnosis. Of course, for colorectal cancer this hypothesis would be unconvincing in the era of screening, when the vast majority of cancers in the general population are diagnosed in early stage, but it is worth noting that the included studies on cirrhosis and synchronous metastases included cancers diagnosed before the implementation of colorectal cancer screening programs.

The issue of potential confounders, such as tumor stage, nodal involvement, location of the primitive cancer (in colorectal cancer), chemotherapy regimen and others, was addressed differently in the included studies. In most studies, the authors chose to perform multivariate analyses to adjust for confounders or analyses on matched subpopulations and only in a few studies were confounders not considered or not clearly defined. Unfortunately, none of the included studies provided the number of events stratified by cancer stage, preventing us from producing stratified estimates. However, the adoption of multivariate analyses in most studies, including adjustment for tumoral stage, should reassure us that no biases were introduced by this potential confounder. Neoadjuvant or adjuvant chemotherapy, on the other hand, was rarely taken into consideration in adjusted analyses. However, the rationale for adjusting for chemotherapy is questionable, since tolerability of chemotherapy could be one of the mechanisms leading to a different occurrence of metastases.

Another concern may derive from the long time period over which the studies were carried out, at least for studies on synchronous metastases. However, differences in results by study period are impossible to distinguish from the effect of the study size. Although the diagnostic methods used for the diffuse liver diseases considered have not varied much during the time interval of our analysis, the included studies adopted different definitions of liver disease even for the same kind of liver injury. Instead, new methods for liver metastasis diagnosis were introduced into clinical practice during the study period. However, this should not affect the results of the metanalysis, since in every study exposed and non-exposed patients were enrolled during the same time span.

Only two small studies on viral hepatitis allowed a comparison of the cumulative occurrence of metastases from diagnosis to follow-up in an unbiased population of patients (Qian 2014 [31]; Zeng 2013 [33]). This comparison does not suggest that differences in synchronous or metachronous metastases may be due to a differentiated assessment of prevalent metastases in the two groups. In fact, both studies found fewer metastases at diagnosis and during follow-up in patients with than in patients without viral hepatitis. Unfortunately, no study allowed this comparison for steatosis or cirrhosis.

Previous metanalyses have reported a lower risk of liver metastases in injured livers. However, they focused on only one type of diffuse liver disease (cirrhosis) or on only one type of primary cancer site (colorectal cancer) [39]. Compared with previous metanalyses, our study pays more attention to the issue of synchronous and metachronous liver metastases, both by conducting clearly separated analyses and by trying to consider them together for the studies that allowed it, since the two phenomena may be communicating vessels.

According to our results, the previously hypothesized mechanisms for a lower risk of liver metastases in injured liver seem to be only partially confirmed, possibly more for viral hepatitis than for other diffuse liver diseases. These mechanisms include the higher concentration of metalloproteinase inhibitor [40], decreased neovascularization [41] and changes in liver-related immunity [42]. All these modifications can affect the various steps required for the transport, implant and adaptation of cancer cells to another organ [43]. Their combined effect may produce a decrease in the effectiveness of the process of metastatization, which is per se inefficient [44].

This systematic review has some limitations. Firstly, since we excluded papers in Chinese and in Japanese and liver disease prevalence is particularly high in Asia, resulting in particularly extensive literature from these countries, we may have missed some relevant papers. Moreover, we considered collectively studies with different assessments of the diffuse liver disease. Liver steatosis can be diagnosed either by imaging techniques or by pathological examination, while for viral hepatitis, some studies included only active disease and others included any serological signs of present or past infection. Only one study (Li 2019 [38]) stratified for different types of viral infection status, suggesting that inactive or resolved infection but not chronic infection had a protective effect on liver metastasis occurrence. Hence, this inconsistency in exposure definitions is probably one of the main sources of residual heterogeneity.

## 5. Conclusions

Diffuse liver diseases seem to be protective against synchronous liver metastases. This could be the result of earlier cancer diagnosis due to opportunistic screening in patients treated and followed up for liver diseases.

A slight protective effect was also found on metachronous liver metastases for viral hepatitis, while the presence of other diffuse liver diseases had no effect or resulted in a slight increase in the risk of developing metachronous liver metastases.

To clearly answer the question of whether and which diffuse liver diseases influence the probability of developing liver metastases, future studies should be population-based, including all incident cases from stage II (at least) to IV and they should assess simultaneously, but separately, the prevalence of synchronous metastases and the incidence of metachronous metastases.

## Figures and Tables

**Figure 1 cancers-13-02246-f001:**
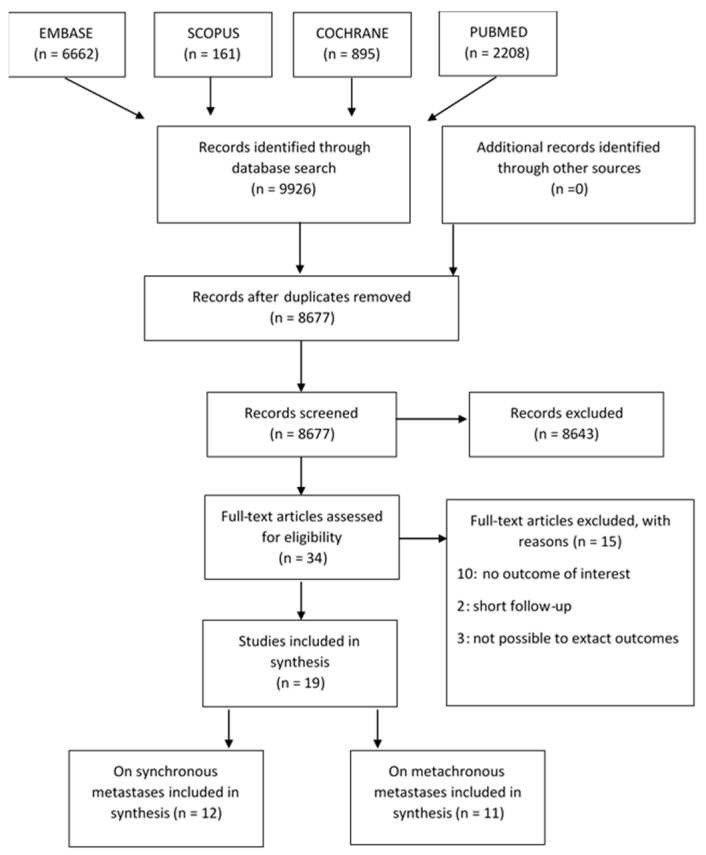
PRISMA flow diagram representing records identified through the literature search, screened and included in the synthesis.

**Figure 2 cancers-13-02246-f002:**
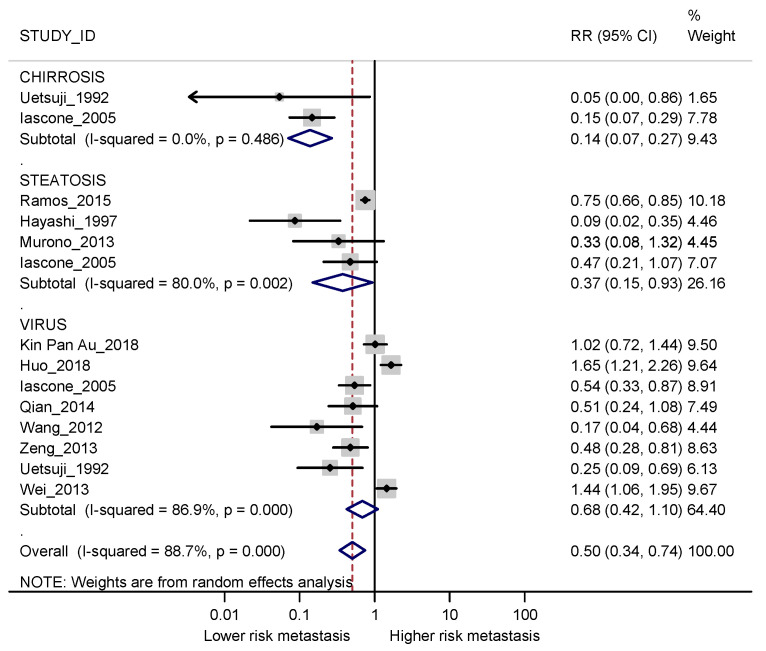
Forest plot displaying the RRs and corresponding 95% CI for synchronous metastases in exposed vs non-exposed patients for different diffuse liver diseases. I2 is reported as a measure of heterogeneity. Overall RR and I2 were calculated after excluding the comparison between patients with and without steatosis for Iascone 2005 [22] since this was the least represented liver disease in the study.

**Figure 3 cancers-13-02246-f003:**
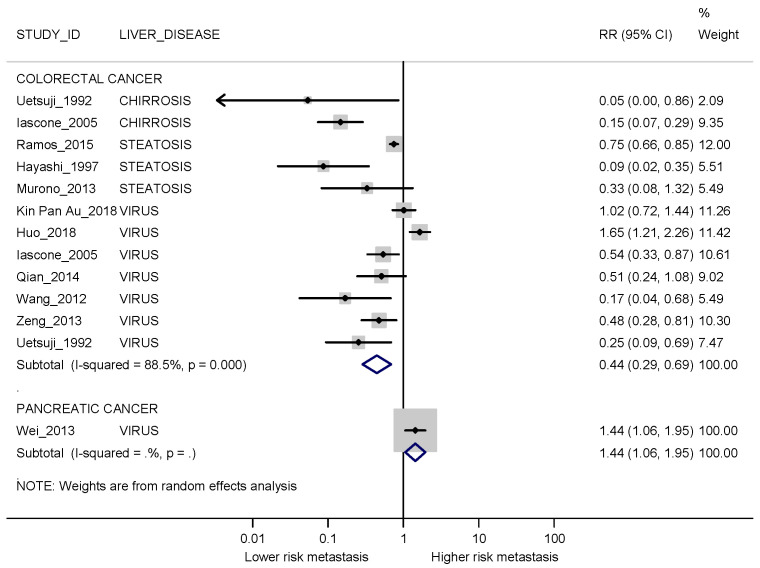
Forest plot displaying the RRs and corresponding 95% CI for synchronous metastases in exposed vs non-exposed patients stratified by cancer sites. I2 is reported as a measure of heterogeneity. The comparison between patients with and without steatosis for Iascone 2005 [22] was not considered since this was the least represented liver disease in the study.

**Figure 4 cancers-13-02246-f004:**
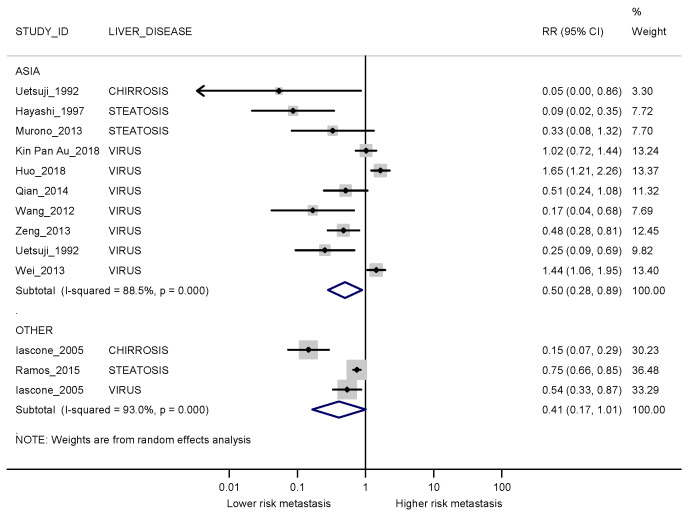
Forest plot displaying the RRs and corresponding 95% CI for synchronous metastases in exposed vs non-exposed patients stratified by country. I2 is reported as a measure of heterogeneity. The comparison between patients with and without steatosis for Iascone 2005 [22] was not considered since this was the least represented liver disease in the study.

**Figure 5 cancers-13-02246-f005:**
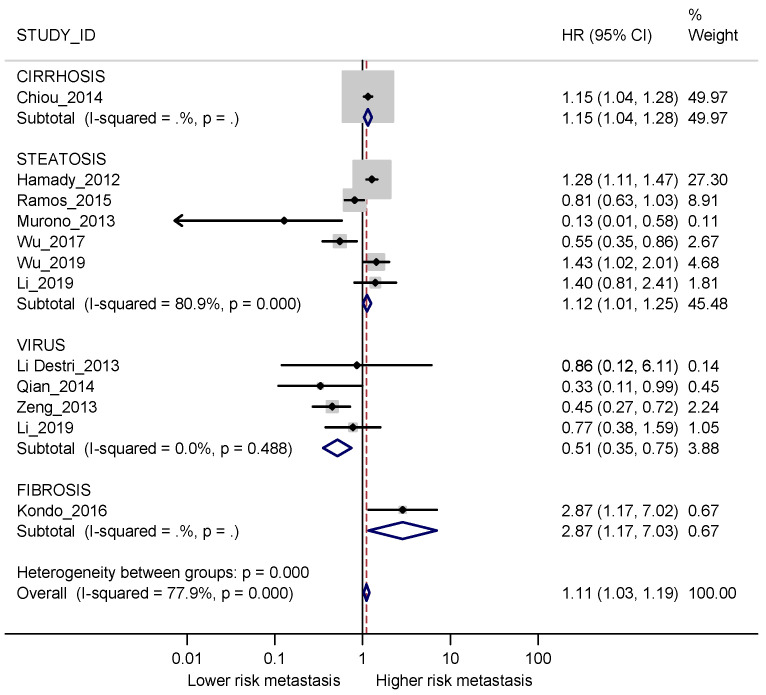
Forest plot displaying the HRs and corresponding 95% CI for metachronous metastases in exposed vs non-exposed patients for different diffuse liver diseases. I2 is reported as a measure of heterogeneity. Overall RR and I2 were calculated after excluding the comparison between patients with and without steatosis for Li 2019 [38] since this was a secondary outcome in the study.

**Figure 6 cancers-13-02246-f006:**
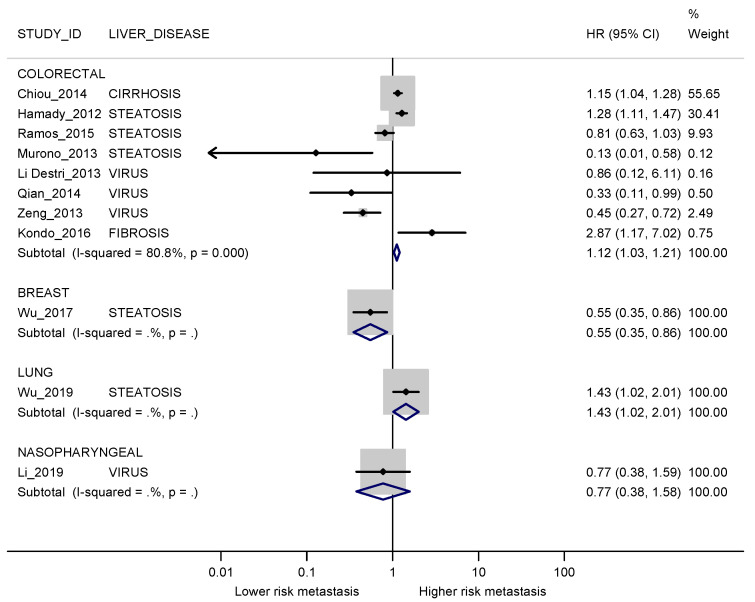
Forest plot displaying the HRs and corresponding 95% CI for metachronous metastases in exposed vs non-exposed patients stratified by cancer site. I2 is reported as a measure of heterogeneity. The comparison between patients with and without steatosis for Li 2019 [38] was not considered since this was a secondary outcome in the study.

**Figure 7 cancers-13-02246-f007:**
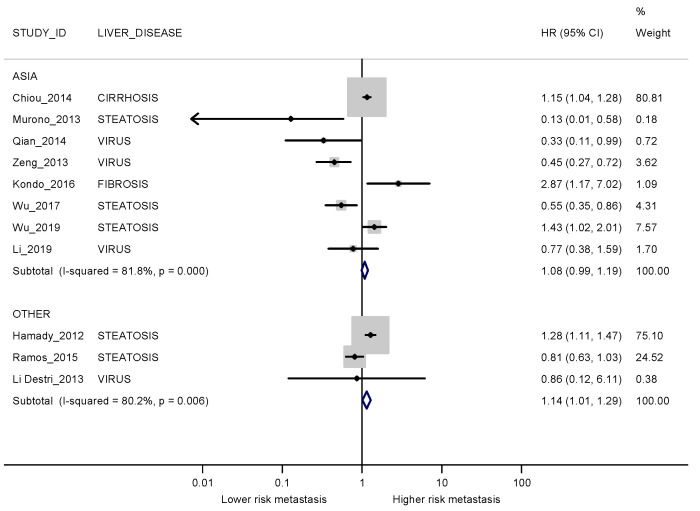
Forest plot displaying the HRs and corresponding 95% CI for metachronous metastases in exposed vs non-exposed patients stratified by country. I2 is reported as a measure of heterogeneity. The comparison between patients with and without steatosis for Li 2019 [38] is not considered since this was a secondary outcome in the study.

**Figure 8 cancers-13-02246-f008:**
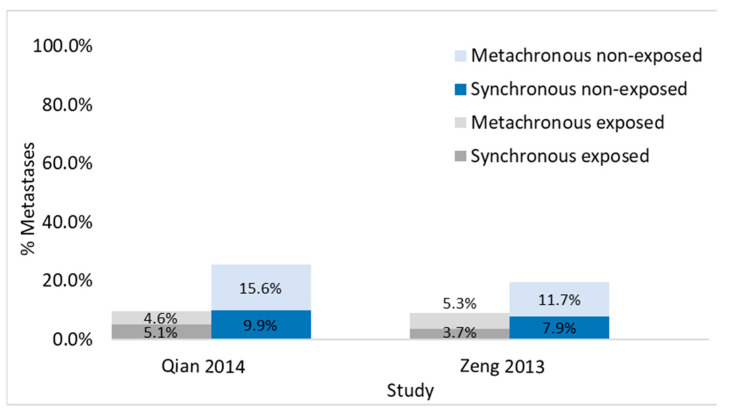
Graph representing the percentages of synchronous and metachronous liver metastases occurring in exposed and non-exposed patients in studies evaluating both types of metastases.

**Figure 9 cancers-13-02246-f009:**
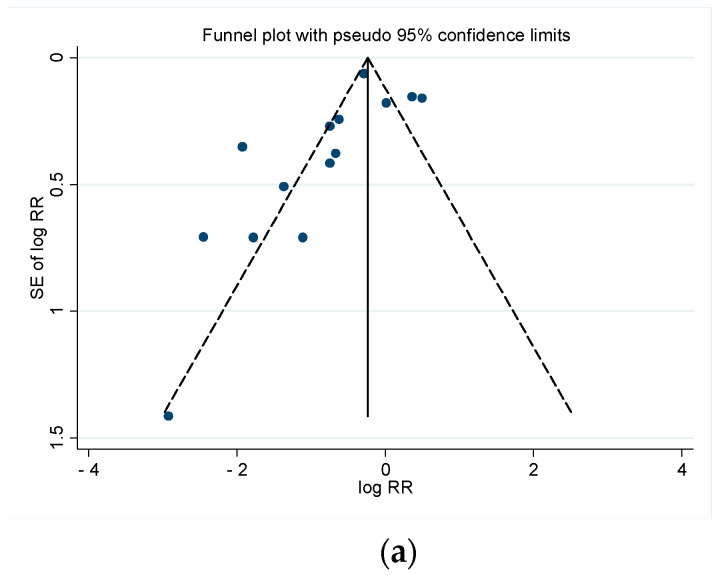
Funnel plots with pseudo 95% confidence intervals for studies on synchronous (**a**) and metachronous (**b**) liver metastases.

**Table 1 cancers-13-02246-t001:** Synopsis of the included studies.

Author/Year	Primary Cancer Site	Diffuse Liver Disease	Study Design	Outcome	Primary/R0	Hospital/Population-Based	Events/ Total	Events/ Exposed	Events/ Non-Exposed	LD Diagnosis Method	LD Diagnosis Timing	Follow-Up Exposed	Follow-Up Non-Exposed	Age (Years) Exposed	Age (Years) Non-Exposed
Uetsuji 1992 [21]	CRC	Cirr	Cr-Sec	S	PR	HOS	40/250	0/46	40/204	Bilirubin > 1.5, Albumin < 3.5, Cholinesterase< 3500, GGT > 46, IG retention rate at 15′ > 15%	At PR surgery	-	-	-	-
Iascone 2005 [22]	CRC	Cirrs	Cr-Sec	S	PR	HOS	182/576	8/171	174/405	Biopsy	At PR surgery	-	-	71.2	65.8
Iascone 2005 [22]	CRC	Steat	Cr-Sec	S	PR	HOS	179/576	5/33	174/543	Biopsy	At PR surgery	-	-	-	-
Uetsuji 1992 [21]	CRC	Vir	Cr-Sec	S	PR	HOS	40/250	4/76	36/174	Blood Test	At PR surgery	-	-	-	-
Huo 2018 [28]	CRC	Vir	Cr-Sec	S	PR	HOS	364/4033	38/244	326/3789	Blood Test	At PR diagnosis	-	-	57.3	62.7
Iascone 2005 [29]	CRC	Vir	Cr-Sec	S	PR	HOS	189/630	15/87	174/543	Blood Test	Before PR surgery	-	-	69.8	65.8
Wang 2012 [32]	CRC	Vir	Cr-Sec	S	PR	HOS	50/354	2/70	48/284	Blood Test	-	-	-	52.9	56.2
Chiou 2014 [20]	CRC	Cirr	Cohort	M	PR	POP	2529/14865	516/2973	2013/11892	NatReg: ICD-9-CM: 571.2, 571.5, 571.6	Before PR surgery	-	-	67.6	67.6
Hamady 2012 [23]	CRC	Steat	Cohort	M	R0	HOS	1118/2715	437/927	681/1788	Biopsy	At LM resection	34		16% > 75	15% > 75
Ramos 2015 [24]	CRC	Steat	Cohort	S/M	R0	HOS	S:511/943	S:194/421	S:317/513	Biopsy	At LM resection	47.05		62.6	62.9
							M:253/528	M:127/264	M:126/264						
Hayashi 1997 [25]	CRC	Steat	Cohort	S	PR	HOS	117/839	2/121	115/718	Ultrasound	At PR diagnosis	-	-	58.5	61.4
Murono 2013 [26]	CRC	Steat	Cohort	S/M	PR	HOS	S:54/604	S:2/63	S:52/541	CT (HU liver/spleen < 1.1)	Before PR surgery	-	-	65	67.2
							M:59/529	M:1/59	M:58/470						
Kin Pan Au 2018 [27]	CRC	Vir	Cohort	S	R0	HOS	185/304	13/21	172/283	Blood Test	Before LM resection	-	-	61	60
Li Destri 2013 [30]	CRC	Vir	Cohort	M	PR	HOS	44/488	1/31	43/457	Blood Test	Before PR diagnosis	108		61	66
Qian 2014 [31]	CRC	Vir	Cohort	S/M	PR	HOS	S:133/1413	S:7/138	S:126/1275	Blood Test	Before PR surgery	72.3		58.5	59.2
							M:185/1150	M:6/131	M:179/1149						
Zeng 2013 [33]	CRC	Vir	Cohort	S/M	PR	HOS	S:211/2868	S:14/373	S:197/2495	Blood Test	-	65		57	61
							M:287/2868	M:19/373	M:268/2495						
Kondo 2016 [34]	CRC	Fibr	Cohort	M	PR	HOS	54/953	8/77	46/876	NFS > 0.676	-	51.2		75.3	64.9
Wu 2017 [35]	Breast	Steat	Cohort	M	PR	HOS	123/1230	27/372	96/858	Ultrasound	At PR diagnosis	30.7	32.4	50% > 50	35% > 50
Wu 2019 [36]	Lung	Steat	Cohort	M	PR	HOS	166/1873	58/408	108/1465	Ultrasound	At PR diagnosis	14.5		67% > 60	51% > 60
Wei 2013 [37]	Pancreas	Vir	Cohort	S	PR	HOS	156/460	29/63	127/397	Blood Test	At PR diagnosis	12		-	-
Li 2019 [38]	NPC	Vir/Steat	Cohort	M	PR	HOS	64/1367	13/123	51/492	Blood Test	At PR diagnosis	27.8		10%≥ 60	11%≥ 60
TOT CRC							32661	5872	26780						
TOT NON-CRC							4930	966	3212						
TOT							37591	6868	29992						

Synopsis of the included studies subdivided by type of diffuse liver disease and reporting primary cancer site, considered outcomes, number of patients included and number of patients in exposed and non-exposed groups. Primitive/R0 refers to patients’ condition at the moment of inclusion (at diagnosis of primitive cancer or after R0 liver metastasis resection). Follow-up is expressed in months. Cirr, Cirrhosis; Steat, Steatosis; Vir, Virus; Fibr, Fibrosis; Cr-Sec, Cross-Sectional; HOS, Hospital-based study; Pop, Population-based study; PR, Primary; CRC, colorectal cancer; NPC, nasopharyngeal cancer; M, metachronous liver metastases; S, synchronous liver metastases; NatReg, National register; TOT, total number of patients; NFS, NAFLD fibrosis score; LM, liver metastasis, HU, Hounsfield Unit; GGT, Gamma-glutamyltransferase; IG, Indocyanine green; ICD, International Classification of Diseases. ICD-9-CM codes: 571.2, alcoholic cirrhosis of liver; 571.5, cirrhosis of liver without mention of alcohol; 571.6, biliary cirrhosis.

**Table 2 cancers-13-02246-t002:** Newcastle-Ottawa Scale appraisal of included studies.

Author/Year	ROB	Selection	Representativeness of EXPOSED Cohort	Selection of Non-Exposed Cohort	Ascertainment of Exposure	Outcome of Interest Not Present at Study Start	Comparability	Comparability on the Basis of the Design or Analysis	Outcome	Assessment of Outcome	Follow-Up Long Enough for Outcomes	Adequacy of Follow-Up
Chiou 2014 [20]	5/9	3/4	☼	☼	☼		1/2	☼	1/3	☼		
Uetsuji 1992 [21]	4/6	3/3	☼	☼	☼		1/2	☼	0/1			
Iascone 2005 [22]	4/6	3/3	☼	☼	☼		1/2	☼	0/1			
Hamady 2012 [23]	8/9	4/4	☼	☼	☼	☼	2/2	☼☼	2/3	☼	☼	
Ramos 2015 [24]	8/9	4/4	☼	☼	☼	☼	2/2	☼☼	2/3	☼	☼	
Hayashi 1997 [25]	7/9	3/4	☼	☼	☼		1/2	☼	3/3	☼	☼	☼
Murono 2013 [26]	6/9	4/4	☼	☼	☼	☼	2/2	☼☼	0/3			
Kin Pan Au 2018 [27]	9/9	4/4	☼	☼	☼	☼	2/2	☼☼	3/3	☼	☼	☼
Huo 2018 [28]	6/6	3/3	☼	☼	☼		2/2	☼☼	1/1	☼		
Iascone 2005 [29]	4/6	3/3	☼	☼	☼		1/2	☼	1/1	☼		
Li Destri 2013 [30]	7/9	3/4	☼	☼	☼		1/2	☼	3/3	☼	☼	☼
Qian 2014 [31]	6/9	3/4	☼	☼	☼		0/2		3/3	☼	☼	☼
Wang 2012 [32]	4/6	2/3	☼	☼			1/2	☼	1/1	☼		
Zeng 2013 [33]	6/9	3/4	☼	☼	☼		2/2	☼☼	1/3	☼		
Kondo 2016 [34]	3/9	1/4			☼		0/2		2/3	☼	☼	
Wu 2017 [35]	8/9	4/4	☼	☼	☼	☼	1/2	☼	3/3	☼	☼	☼
Wu 2019 [36]	7/9	4/4	☼	☼	☼	☼	1/2	☼	2/3	☼		☼
Wei 2013 [37]	6/9	4/4	☼	☼	☼	☼	0/2		2/3	☼		☼
Li 2019 [38]	9/9	4/4	☼	☼	☼	☼	2/2	☼☼	3/3	☼	☼	☼

Appraisal of included guidelines using NOS for cohort and case-control studies and modified NOS for cross-sectional studies. T, primary cancer site; CRC, colorectal cancer; NPC, nasopharyngeal cancer; MET, metachronous liver metastases; SYN, synchronous liver metastases.

**Table 3 cancers-13-02246-t003:** Overall survival in exposed and non-exposed patients.

Study	5-Year Survival Exposed	Median Survival Exposed	5-Year Survival Non-Exposed	Median Survival Non-Exposed
Hamady 2012 [23]	39.3%	22 months	42.8%	24 months
Ramos 2015 [24]	55.1%		45.2%	
Qian 2014 [31]	40.6%		39.3%	
Zeng 2013 [33]		56 months		49 months
Kondo 2016 [34]	79.8%		92.2%

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
