# Peer review of "The Effect of Diffuse Liver Diseases on the Occurrence of Liver Metastases in Cancer Patients: A Systematic Review and Meta-Analysis"

_cancers, 2021, doi:10.3390/cancers13092246_

Round 1
Reviewer 1 Report
This systematic review and meta-analysis addresses the issue of the effect of diffuse liver disease on synchronous and metachronous liver metastases from cancer of various origin.
The topic is relevant and original. The study is well-written, conducted and discussed
MINOR REVISIONS
In the abstract Metachronous has been abbreviated with “–L” instead of “-M”.
Author Response
We thank the Reviewer for the overall positive judgement of our work.
We have changed the abbreviation in the abstract from L- to M-.
Reviewer 2 Report
In their review and meta-analysis, the authors have made a thorough and extensive study on the effect of diffuse liver disease on the occurrence of metachronous and synchronous liver metastasis in solid tumors. I have only one comment and the others are minor.
Main comment:
in the discussion section paragraph 8, the authors briefly mention the mechanism behind the protective effect of diffuse liver disease particularly viral hepatitis on liver metastasis which is their main finding. I would therefore advise to discuss the protective mechanisms in more detail in order to back up the findings of their review.
minor comments:
In the abstract, the authors use (L) abbreviation for metachronous liver metastasis whereas it should be (M)
In section 3.1 of the results, paragraph 2 “while four focused on other cancer, breast cancer, non-small cell lung cancer, pancreatic cancer” please use “other cancers”
In section 3.3.3. “Two studies permitted an evaluation of the cumulative incidence of both synchro[1]nous ad metachronous liver metastases on the same patient groups, with and without” please fix "ad" to "and".
Author Response
We thank the Reviewer for the overall positive judgement of our work. See below for the point-by-point Responses.
Main comment:
in the discussion section paragraph 8, the authors briefly mention the mechanism behind the protective effect of diffuse liver disease particularly viral hepatitis on liver metastasis which is their main finding. I would therefore advise to discuss the protective mechanisms in more detail in order to back up the findings of their review.
RE: We thank the Reviewer for the comment. We have added two sentences with the respective references to the paragraph:
"These mechanisms include the higher concentration of metalloproteinase inhibitor [40], decreased neovascularization [41], and changes in liver-related immunity [42]. All these modifications can affect the various steps required for the transport, implant, and adaptation of cancer cells to another organ [43]. Their combined effect may produce a decrease in the effectiveness of the process of metastatization, which is per se inefficient [44]."
minor comments:
In the abstract, the authors use (L) abbreviation for metachronous liver metastasis whereas it should be (M)
RE: We thank the Reviewer and we have changed the abbreviation.
In section 3.1 of the results, paragraph 2 “while four focused on other cancer, breast cancer, non-small cell lung cancer, pancreatic cancer” please use “other cancers”
RE: We thank the Reviewer and we have changed the text accordingly.
In section 3.3.3. “Two studies permitted an evaluation of the cumulative incidence of both synchro[1]nous ad metachronous liver metastases on the same patient groups, with and without” please fix "ad" to "and".
RE: We thank the Reviewer and we have fixed the text.
Reviewer 3 Report
Interesting topic. Appropriate research study. Well presented results.
Good job.
Author Response
We thank the Reviewer for the positive judgement of our work.